# Task-Oriented Hierarchical Object Decomposition for Visuomotor Control

**Jianing Qian[1], Yunshuang Li[2], Bernadette Bucher[3], Dinesh Jayaraman[1]**
University of Pennsylvania[1], University of Southern California[2], University of Michigan[3]
{jianingq,dineshj}@seas.upenn.edu, yunshuan@usc.edu, bucherb@umich.edu

**Abstract:** Good pre-trained visual representations could enable robots to learn visuomotor policy efficiently. Still, existing representations take a one-size-fits-all-tasks approach that comes with two important drawbacks: (1) Being completely task-agnostic, these representations cannot effectively ignore any task-irrelevant information in the scene, and (2) They often lack the representational capacity to handle unconstrained/complex real-world scenes. Instead, we propose to train a large combinatorial family of representations organized by scene entities: objects and object parts. This hierarchical object decomposition for task-oriented representations (HODOR) permits selectively assembling different representations specific to each task while scaling in representational capacity with the complexity of the scene and the task. In our experiments, we find that HODOR outperforms prior pre-trained representations, both scene vector representations and object-centric representations, for sample-efficient imitation learning across 5 simulated and 5 real-world manipulation tasks. We further find that the invariances captured in HODOR are inherited into downstream policies, which can robustly generalize to out-of-distribution test conditions, permitting zero-shot skill chaining. Appendix, code and videos: https://sites.google.com/view/hodor-corl24.

**Keywords:** Visual Representations, Entities, Imitation, Manipulation

## 1 Introduction

How should an embodied agent internally represent its visual observations? The role of a representation is to supply *task-relevant information* to the decision-making procedure, such as a neural network policy. This implies two desirable properties for any representation. **First**, a representation should highlight task-relevant aspects of the scene and de-emphasize irrelevant information. This means that good representations are inherently task-specific: driving on the road relies on very different aspects of the scene than walking on the pavement. Or, in our experimental settings, the same kitchen robot might need to attend to different scene aspects when executing different sub-tasks: boiling water, turning on a faucet, putting vegetables into a pot, and so on. This necessary task-specificity was an important motivation for "end-to-end" approaches in the last decade [1, 2]. Rather than hand-crafted "off-the-shelf" representations (e.g., bags of visual words [3], SIFT [4], HoG features [5]), they aimed to optimize the entire control pipeline for one or a few tasks. **Second**, representations must *conveniently* organize scene information so that the policy can be represented in a simple function. For robot learning, such simplicity in optimal policies translates to ease of discovering these policies by an optimization procedure operating over limited experience. For example, a representation could aim to *disentangle* the latent factors of variation in the inputs [6, 7, 8, 9, 10].

In recent years, motivated by efforts to improve data efficiency, off-the-shelf representations have risen in popularity again, this time in the form of the activations of pre-trained vision models [11, 12, 13, 14, 15, 16, 17, 18]. While these pre-trained representations are no longer hand-crafted, they retain another critical drawback of off-the-shelf representations: they are agnostic to the downstream

8th Conference on Robot Learning (CoRL 2024), Munich, Germany.

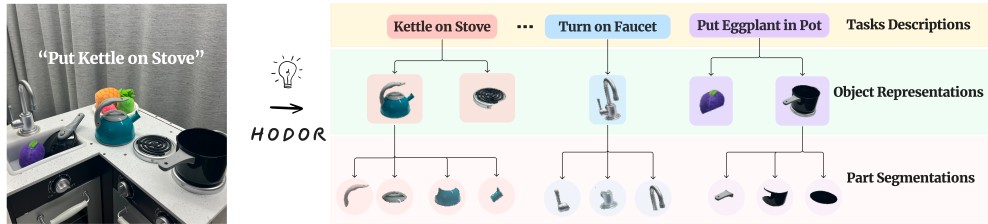

Figure 1: HODOR embeds a 2D input image into an object-centric, multi-resolution, task-specific representation for efficient policy learning

task. As such, they are trained to capture all scene information, even though most of this information might be irrelevant to any given downstream use case. Furthermore, in complex, cluttered scenes, these standard fixed-dimensionality "scene vector" representations are limited in their ability to comprehensively capture all fine-grained details.

How might we enable task specificity alongside representation pre-training? Rather than train a single representation, we propose to generate a menu of representations that can be combinatorially assembled into the right platters suited for each downstream task. In particular, we propose hierarchical object decomposition for task-oriented representations (HODOR). HODOR recognizes that *scene entity trees*, i.e., trees of objects and object parts, provide a convenient organizing principle for a representation menu: different objects are relevant at different levels of detail to different tasks or task phases. Further, HODOR exploits advances in semantic visual understanding to select task-relevant scene entity trees for any given task. HODOR involves no new pre-training to generate representations; instead, it mines these representations from existing vision and language foundation models. See Fig 1.

Our main contributions are: 1) We introduce a pre-trained approach to generate a multi-level decomposition of a given scene and then "filter" it based on language-based task descriptions into a task-specific tree; 2) we present an approach for downstream policy networks to learn from this, and structured HODOR representation, and 3) We empirically show that HODOR is better than prior pre-trained representations at downstream manipulation tasks.

## 2 Related Work

**Pre-trained representations in computer vision and robotics** Rather than learn from scratch for each new task, pre-trained representations are attractive for their potential to bootstrap task learning, reducing task-specific training data requirements. Pre-trained visual representations have demonstrated their potential utility for many downstream visual recognition tasks [11, 12, 19, 20, 21, 22]. More recently, robot learning researchers applied these pre-trained representations to robotic tasks [23, 13, 14, 15, 24]. Several works have recently studied whether, when, and how these representations improve robot policy learning [25, 26, 27]. The above approaches all focus on what we call "scene vector" representations i.e., summarizing an image in a monolithic, unstructured, fixed-dimensionality vector. HODOR instead structures the representation along scene entities.

**Object-centric embeddings (OCE)** With constrained task setups, roboticists have long used object bounding boxes, poses, and shapes to represent state, usually from bespoke domain-specific supervised models. A recent wave of object-centric representation learning approaches have proposed modified neural network architectures that encouraged self-supervised learning to generate discrete objects and entities [28, 29, 30, 31, 32, 33, 34, 35, 36, 37] [38, 39]. While they have yielded promising results on small domains with large datasets and few objects, it has proven difficult to scale such approaches to train effective object-centric embeddings on in-the-wild images. Meanwhile, advances in foundation models for open-world visual recognition have made it possible once more to assemble object-centric representations for robotic use cases without needing to *train* such representations from scratch [40, 41, 42]. Closely related to our method, POCR [41] is a recent

object-centric scene encoder that leverages the zero-shot segmentation ability of vision foundation models such as SAM [43] and LIV [15]. Like POCR, HODOR also assembles object-centric "off-the-shelf" embeddings but with two important new components: hierarchical scene entity trees and task-conditioned object selectivity.

**Hierarchical coarse-to-fine representations**  Recognizing that different levels of detail are appropriate for different downstream use cases, several methods aim to build multi-resolution scene vector representations within neural networks [44, 45, 46]. Similar to HODOR, others have also represented *objects* at multiple resolutions [47, 48, 49, 34, 50, 51]. In robotics, Ravichandran et al. [50] build 3D scene graphs for representing multi-resolution scenes in a SLAM framework; in this context, they represent full metric-semantic 3D meshes at the levels of "objects", "places", "rooms", and "buildings". Ok et al. [51] represents object shapes for a navigation agent more coarsely for farther away objects and more finely for nearby ones. Liu et al. [52] builds 3D diffusion models for robotic manipulation skills by leveraging object-part decomposition. Geng et al. [53] builds a large-scale 3D manipulation benchmark that enables cross-category generalization by annotating object-part decomposition. In contrast, we work with 2D image observations, construct *task-conditioned* representations, and further decompose objects into parts to facilitate fine-grained manipulation tasks.

**Task conditioning in policies and representations**  For conditioning policies on task identity in multi-task learning settings, most prior works [54, 15] propose to append embedded language-based task descriptions alongside sensory observation embeddings. Rather than filtering out task-irrelevant information within the representations input to the policy, such approaches rely on the downstream policy to separate relevant from irrelevant information. More recently, many works suggest that obtaining task-relevant scene representations benefits downstream manipulation tasks. Sundaresan et al. [55], Parashar et al. [56] map task descriptions to interaction points 3D space. Marza et al. [57], Chen et al. [58] propose to process visual observations into task-relevant embeddings before passing them as input to the policy. Ma et al. [59], Eftekhar et al. [60] use task descriptions to identify task-relevant regions in a 2D observation to guide explorations for reinforcement learning. Zhu et al. [40] ask users to scribble over task-relevant objects on image observations to build task-relevant scene representations. With HODOR, we build scene representations tied to entity trees in the scene, which are automatically filtered into task-relevant representations by selecting the appropriate entities as dictated by task semantics.

To sum up, HODOR presents a new visual representation framework for robotics that is object-centric, task-conditioned, and multi-resolution. In our systematic experimental study, we find that the combination of these properties in HODOR yields important gains in task performance across a variety of robot manipulation tasks.

## 3  Inferring HODOR Representations From Foundation Models

As motivated above, we propose to assemble image representations that are object-centric, task-conditioned, and multi-resolution. Below, we describe the step-by-step procedure for inferring the HODOR representation for an input image.

### 3.1  Preliminaries: Object-Centric Image Embeddings from Pre-trained Foundation Models

Following common convention [28, 29, 30, 31, 32, 33, 34, 35, 36, 37], we define an object-centric embedding (OCE) $o^t$ of an image observation at time step $t$ as a set of "slots" $s_i^t$ that summarize the entities (e.g., objects, parts, sub-parts) in the image. Each entity slot $s_i^t = (l_i^t, z_i^t)$ containing of its location $l_i^t \in \mathbb{R}^4$ (e.g. axis-aligned bounding boxes denoted by four coordinates) and content embedding $z_i^t \in \mathbb{R}^D$.

We start from successful recent OCE approaches [40, 41] that leverage the zero-shot segmentation ability of vision foundation models such as SAM [43]. Specifically, we follow Zhu et al. [40] to

obtain SAM object segment "slot" masks $m^t = \{m_i^t\}$ on the first frame $t = 0$ of a policy rollout or demonstration, and then track each mask for the rest of the rollout using XMem [61].

Now that we have a set of entity masks $m_t$ of the scene, how can we assemble them into a hierarchical structure that best describes the task at hand? In Section 3.2, we describe our approach that selects task-relevant entities conditioning on the task descriptions. In Section 3.3, we describe our algorithm to discover and group parts into the task-relevant objects, and finally, in Section 4, we describe our policy design to best utilize our representations for downstream robotics tasks.

## 3.2 Task-Conditioned Entity Selection

Consider a robot in a kitchen environment, as in our experiments in Sec 5. This environment has a large task space, including boiling water, putting vegetables in a pot, or turning on the faucet. Each task requires the robot to pay attention to different parts of the scene. Most prior approaches naively use a constant scene representation for all tasks. Instead, we propose to construct bespoke task-relevant representations. We exploit recent advances in vision and language foundation models to permit specifying task-relevant parts of the scene using natural language task descriptions. Given the task description $d$, we queried large language models (GPT-4 [62]) to identify the entities in $d$. For example, the task-relevant object for "open the microwave" is "microwave." After identifying the noun entities, we prompt Grounded SAM [63], an open-set object segmentation model with these entity names, to produce a set of task-relevant segmentation masks $M_{tr} = \{m_i\}$.

## 3.3 Hierarchical Object Decomposition

Given the image observation $I$ and the selected task-relevant object masks $M_{tr}$, we can assemble the hierarchical OCE. For our settings, we set up three levels in the hierarchy. The first level consists of the root node, which is the scene-level slot $s_{scene} = (l_{scene}, z_{scene})$. Similar to [16], we use the CLS token of DINO-v2 ViT as the scene-level representations $z_{scene}$. We define $l_{scene} = [0., 1., 0., 1.]$ in normalized scene coordinates, representing a bounding box containing the entire image. The second level contains an ordered sequence of task relevant object slots $[s_{i,obj}] = [(l_{i,obj}, z_{i,obj})]$. $z_{i,obj}$ is computed by average-pooling DINO-v2 features of the last attention layer over the corresponding segmentation mask $m_i \in M_{tr}$, and $l_{i,obj}$ is the tightest bounding box around $m_i$.

Finally, the third level contains an unordered set of part slots $\{s_{j,part}\} = \{(l_{j,part}, z_{j,part})\}$ corresponding to each parent object. To discover the object parts, we input image observation $I$ into SAM to obtain all the segmentation masks at different levels of granularity. We assign a part $j$ to an object parent $i$ if their intersection is greater than 0.5 with an object $i$. We calculate the slot vectors of parts $z_{j,part}$ in the same way as for the objects.

The resulting HODOR representation $[s_{scene}, [s_{i,obj}], [\{s_{j,part}\}_i]]$, contains three layers of slots in coarse-to-fine order: the full scene, task-relevant objects, and their corresponding object parts. HODOR conveniently organizes the scene information into entities and limits the finer levels of the representation to directly task-relevant objects, while still retaining sufficient coarse information to, say, avoid collisions with the rest of the scene. Further, as the scene and the task grow in complexity, the size of HODOR representations grows in tandem to accommodate the increased policy needs for task-relevant scene information. Finally, this representation involves redundancies that may improve robustness to failures in object segmentation or tracking: coarse details of task-relevant objects are still retained in the scene-level slot.

# 4 Policy Architecture and Training

We have seen above that HODOR produces a structured representation containing nested ordered sequences and unordered sets of slots. For the policy to flexibly process this representation, we use a transformer-based policy architecture, illustrated in Fig 2.The input self-attention layer naturally processes variable-sized inputs as well as unordered sets without any special modifications. We feed HODOR representations into the self-attention layer as follows. Each individual slot at each level

of HODOR is represented as an input token. To represent the level-wise ordering, we attach a level embedding to the slot vectors. Similar to positional encodings in common transformer architectures, these level embeddings $\phi$ are learnable tokens that are designed to encode hierarchical graph information. We learn one token $\phi_{scene}$ for the scene and one token $\phi_{obj(i)}$ for each object. All parts $\{s_{j,part}\}_i$ that belongs to the same parent object $i$ share the same embedding $\phi_{part(i)}$. Like DINO-v2, we append a learnable CLS token to the input so that the self-attention output can accommodate an input summary in the output CLS token. We can either feed the CLS token directly into the downstream MLP or take in the concatenation of all tokens output from the attention layer. More details are in the Appendix A.1. In our experiments, we train the above policy architecture on expert demonstrations with a standard behavior cloning objective for each target manipulation task.

## 5    Experiments

Our method assembles task-oriented entity-based scene representations for manipulation tasks. With simulated and real robot experiments, we aim to answer: **(1)** Do imitation learned policies for robot manipulation tasks benefit from HODOR compared to alternative pre-trained visual representations? **(2)** Do policies trained on HODOR representations acquire more invariance to task-irrelevant information compared to others? **(3)** To what extent are HODOR's gains attributable to its task-oriented, multi-resolution, and object-centric properties?

### 5.1    Simulated Experiments

In order to evaluate the performance of our hierarchical scene representations, we first test our method and baselines on five simulated Franka Kitchen [64] tasks: SlideCabinetDoor, OpenMicrowave, OpenCabinetDoor, TurnOnLight and Turn-Knob. Fig. 4 shows these tasks. The 9-DoF Franka robot is controlled through joint velocities. Following the representation evaluation paradigm in [16], we collected 40 demonstrations for each task and evaluated the behavioral cloning (BC) policy performance as a function of varying numbers of demonstrations. Each method is trained with 3 random seeds. We report the success rate in Fig. 5, which is calculated on 100 online rollouts, and we take the maximum success rate over the course of training. In practice, prompting Grounded SAM with the object name in the simulated environment does not reliably identify the objects due to the fact that objects in the simulated environment could look ambiguous. Instead, we simplify this by drawing tight bounding boxes around target

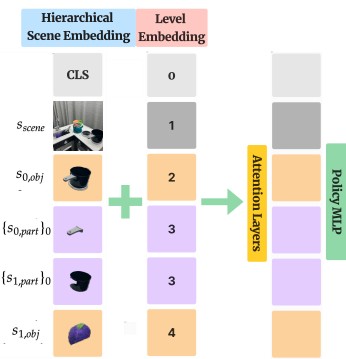

Figure 2: Policy Architecture. We illustrated the resulting HODOR representations for task `Put Eggplant in Pot`. There are two relevant objects in this task: eggplant and pot. Their corresponding parts are also visualized.

objects. In our experiments, we hand-annotate task-relevant object locations for TurnKnob, TurnOn-Light, and OpenCabinetDoor; more details are provided in the Appendix A.2.

**Baselines:** We compare HODOR against four prior pre-trained image representations for robots: **DINOv2** [12] representations are widely used for downstream vision and robotics tasks; similar to [41], we use the last-layer CLS token. **R3M** [13] and **LIV** [15] are popular robotics-targeted scene vector representations. Finally, as described in Sec. 2. **POCR** [41] builds 1-level *object-centric* representations using SAM [43] and LIV.

**Results:** Fig. 5 shows success rates for HODOR and baselines. Please see the videos of the policy rollout in the URL above. R3M and LIV, which are trained on robot-specific data, outperform DINOv2, which is trained on standard vision tasks. HODOR outperforms all these methods with nearly all demonstration set sizes on all tasks besides OpenCabinet-Door, with higher average performance and lower standard errors (shaded region) throughout. The comparison with DINOv2 is particularly instructive since HODOR itself relies on DI-NOv2 encodings of each slot. Here, HODOR is vastly superior, with particularly large gains

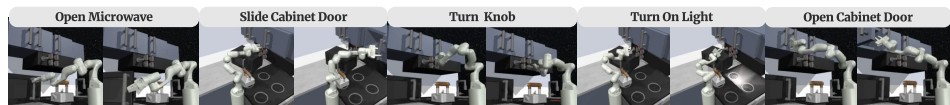

Figure 4: Visualization of simulated Franka Kitchen tasks.

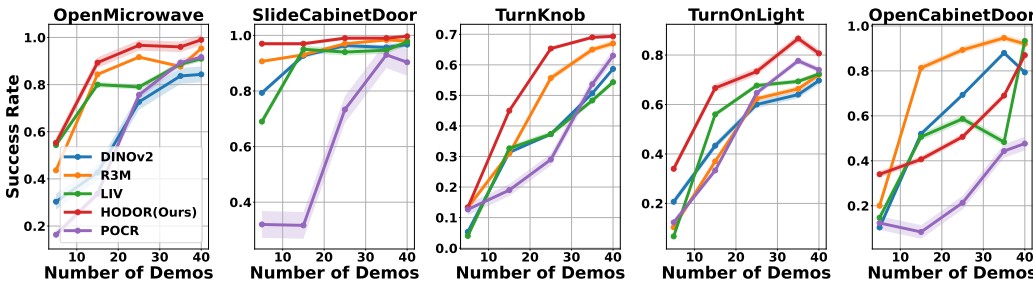

Figure 5: BC performance of HODOR and baselines as a function of the number of demonstrations measured by success rate on five tasks from the FrankaKitchen benchmark. Shaded regions show the standard error across 3 seeds.

in low-data settings. Likewise, POCR, which builds a task-agnostic one-level object-centric representation that includes distractor objects too, and suffers with few demos. We see that HODOR achieves data efficiency benefits from focusing its representation on task-relevant objects.

Overall, these results validate that HODOR's carefully structured representations can overcome the relative limitations of DINOv2 embeddings to match and outperform the best pre-trained image representations, such as R3M and LIV. On one outlier task, OpenCabinetDoor, HODOR does not achieve the best results. This task poses particular difficulties for HODOR because the target objects are not always visible, either occluded by the robot arm or outside the image field of view. This trips up HODOR's object detection and tracking pipeline, affecting representations and policies trained with few demos.

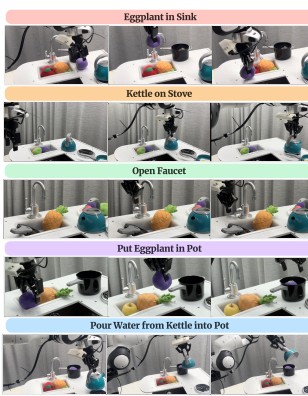

Figure 3: Task Visualization.

**Ablations:** The HODOR framework confers two important new properties to an object-centric representation: multi-resolution and task-orientedness. To what extent do these contribute to HODOR's success? We design two ablation experiments: 1) **Ours−multi-level** is the variant of our method without the object decomposition: instead of the full HODOR representations $[s_{scene}, [s_{i,obj}], [\{s_{j,part}\}_i]]$, we only input scene and object representations (no parts) $[s_{scene}, \{s_{i,obj}\}]$ to the downstream policy. Note that this is different from the POCR baseline above in subtle implementation details, described in Appendix A.9. 2) **Ours−task conditioning** (Ours-TC) is the variant of our method without task conditioning. This means instead of selecting task-relevant objects, we include all objects and their parts in the image representation. Fig. 6 shows these results. In general, Ours−multi-level performs comparable to R3M and LIV, but still worse than HODOR, especially when number of demonstrations is low. This shows that our hierarchically structured representations help agents to learn more efficiently from demonstrations. On the other hand, Ours−task conditioning performs much worse than HODOR. Without task-conditioned entity selection, the representation space grows too large.

## 5.2 Real Robot Experiments

We now evaluate HODOR on a set of 5 *real* robot tasks in a tabletop kitchen manipulation environment. Our tasks involve a variety of skills, such as picking, placing, twisting, and pouring. **Task Design:** We create a kitchen environment with various objects on the countertop. We use a 7-DoF Franka robot arm with continuous joint-control action space. Details of hardware setup are

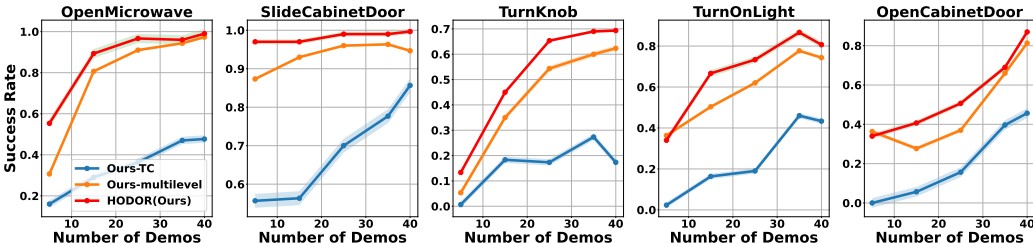

Figure 6: BC performance of HODOR and ablation experiments as a function of the number of demonstrations measured by success rate on two tasks from the FrankaKitchen benchmark. Shaded regions show the standard error across 3 seeds.

| Task Method | Eggplant-Sink | | Kettle-Stove | | Faucet | | Eggplant-Pot | | Water-Pot | | Overall | |
|---|---|---|---|---|---|---|---|---|---|---|---|---|
| # of Trials | 15 | 15 | 15 | 15 | 15 | 15 | 15 | 15 | 15 | 15 | 75 | 75 |
| | IND | OoD | IND | OoD | IND | OoD | IND | OoD | IND | OoD | IND | OoD |
| HODOR (Ours) | **14** | **12** | **13** | 12 | 12 | **8** | **13** | 8 | **12** | **9** | **64** | **49** |
| DINOv2 [12] | 8 | 5 | **13** | 0 | **15** | 5 | 11 | 7 | 11 | 1 | 58 | 18 |
| LIV (ResNet50) [15] | 9 | 1 | 6 | 0 | 8 | 2 | 8 | 0 | 5 | 0 | 36 | 3 |
| R3M (ResNet50) [13] | 6 | 0 | 6 | 0 | 3 | 2 | 10 | 0 | 8 | 0 | 33 | 2 |

Table 1: IND and OoD BC Results on Real Robot Tasks. We report the number of success of each task out of 15 trials.

in the appendix. For each task, we collect 100 demonstrations via teleoperation; for each trajectory, the positions of relevant objects in the scene are randomized within a fixed distribution. We design a suite of tasks that are progressively harder, in terms of manipulation difficulty and complexity of the environment. Fig 3 shows an illustration of each task:

1. Put Eggplant in Sink (`Eggplant-Sink`): This is the easiest task that only involves picking and placing soft plush toys. We create a more challenging visual scenario by cluttering the sink with other fruits.

2. Place Kettle on Stove (`Kettle-Stove`): Pick up the kettle by the handle, and put it onto the stove. The initial position and orientation of the kettle is randomized.

3. Open Faucet (`Faucet`): Twist the knob of the faucet that is placed over the sink. Since the knob is small, it requires the agent to learn precise actions in order to turn it on.

Previous tasks only involve one task-relevant object. To further test the ability of our representation when multiple task-relevant objects are involved, we designed the two following tasks:

4. Put Eggplant in Pot (`Eggplant-Pot`): Pick up the eggplant from the sink and put it in the pot. This task involves two objects: "eggplant" and "pot." The initial positions of both objects are randomized.

5. Pour Water from Kettle into Pot (`Water-Pot`): Pick up the kettle by the handle, move it over to the pot, and pour water out of the kettle (simulated by small orange beads for better visualization) into the black pot. This is by far the hardest task since grasping and pouring out of a heavy kettle requires a firm and secure grasp of its handle. Both the position of the kettle and the pot are randomized.

**Evaluation:** Aside from the graded difficulty of these 5 tasks, we also further study policy performance for each task under an easy "in-distribution" (IND) setting and a hard "out-of-distribution" (OoD) setting. In IND setting, task-irrelevant objects are kept at the same locations as in the collected demonstrations. We report the number of successful runs out of 15 trials for each task. To evaluate the systematic generalization abilities of HODOR, we design three OoD scenes for each task and each OoD scene is evaluated over 5 trials (15 trials in total). In the OoD case,

task-irrelevant objects are either placed at unseen locations or removed. Fig. 7 shows illustrations of the OoD setup.

**Results:** Table 1 shows the results of all methods. Please see URL for videos. HODOR beats all three baselines even in IND settings, and its gains are particularly large in OoD, with LIV and R3M faring particularly poorly. These baselines suffer from both a flat scene vector representation that struggles to capture fine-grained object details needed in our tasks and a one-size-fits-all-tasks approach to building visual representations when many distractor objects are present. To take some examples, `Kettle on Stove` and `Open Faucet` demand fine-grained representation of object parts, such as the kettle handle and faucet knob, and HODOR enjoys the benefit of representing the relevant entity trees. Next, `Put Eggplant in Pot` in the OoD setting involves large distractor objects that are unseen

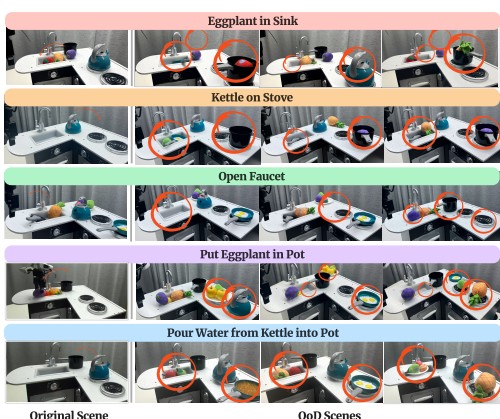

Figure 7: OoD scene settings in real robot experiments. Red circles mark the changes in the OoD scenarios compared to the demonstrations.

during training. HODOR, being inherently invariant to task-irrelevant scene objects, transfers best to such settings. In the last two tasks, `Put Eggplant in Pot` and `Pour Water from Kettle into Pot`, HODOR grows its representation to accommodate multiple task-relevant objects and outperforms baselines.

**Zero-shot Skill Chaining:** To capitalize on our promising OoD results above showing systematic generalization to distractor objects, we attempt to execute the five skills one after another in a pre-determined sequence resembling an eggplant preparation recipe: pick up the kettle and put it on the stove, put the eggplant in the sink, turn on the faucet, put eggplant in pot and finally pick up kettle and pour water into

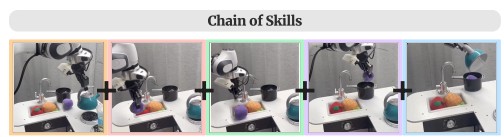

Figure 8: Illustration of zero-shot skill chaining. We demonstrate executing five skills in sequence, creating hard OoD setups.

the pot. Note that each skill at training time is trained from the same starting state distribution, so each of the second, third, fourth, and fifth skills here must handle very far-out-of-domain initial states produced by the previous skills in the sequence. For example, since the eggplant is in the sink while the agent tries to turn on the faucet, it alters the appearance of the scene large enough that our baselines fail to locate the faucet. After many trials, no baseline goes past the second skill in this skill-chaining setting, but HODOR can successfully chain all five, as demonstrated in Fig 8.

## 6 Conclusions and Limitations

We have presented HODOR, a hierarchical task-oriented scene representation generated through object decompositions. Instantiated from vision foundation models, HODOR outperforms state-of-the-art pre-trained representations for robotic policy learning. It also demonstrates robustness to variations in task-irrelevant scene aspects enabling impressive new use cases such as zero-shot chaining of separately trained skills. While these empirical results are impressive, combining multiple pre-trained models creates several potential failure points for HODOR; we analyze these in detail in Appendix A.7. In short, we find that the overall system is often capable of correcting for component failures, and/or such failures are rare in the types of tasks most frequently studied in imitation learning for manipulation today. This is reflected in HODOR's strong results across our extensive evaluations.

**Acknowledgments**

This project was supported in part by NSF CAREER Award 2239301, NSF Award 2331783, and ONR award N00014-22-12677.

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

# A   Supplementary Material

We first go over policy architecture details in Section 4. We then present additional descriptions of the simulated experiment setup in Section 5.1 and the real robot experiment setup in Section 5.2. Lastly, we show results for one additional ablation experiment for the real robot experiments in Section 5.2. For visualization of the real robot experiments as well as the zero-shot skill chaining experiments mentioned in Section 5.2, please see the videos attached.

## A.1   Architecture Implementation Details

In Section 4 of the main paper, we overviewed the architectural choices. Here, we provide a more detailed description of the implementation details. In both simulated and real robot experiments, we train a separate policy for each task. For both environments, we use DINO-v2 with ViT-B/16 backbone to encode objects and parts. In simulated experiments, the policy network is implemented as a 4-layer MLP with hidden sizes [512,256,128], and the concatenation of all token outputs from the attention layer is taken in. In real robot experiments, we use a 3-layer MLP with hidden sizes [1024,1024] instead. Under the robot's hardware constraint, we only input the CLS token into the policy MLP to reduce the number of parameters. All methods share the same policy network architecture.

## A.2   Simulated Experiments Setup

In Section 5.1 of the main paper, we briefly describe the five simulated tasks. Now we will go over a detailed description of each task and how the task-relevant objects are selected: In **OpenMicrowave**, the goal is to open a microwave sitting on a kitchen counter. It requires the agent to locate the microwave and its handle. In **SlideCabinetDoor** and **OpenCabinetDoor**, agents need to locate the handle of cabinet doors and open them. In **TurnOnLight** and **TurnKnob**, agents need to turn the perspective knobs on a panel. In **OpenMicrowave**, the task-relevant object is selected by prompting GroundedSAM with "microwave." In **SlideCabinetDoor**, the task-relevant object is selected by prompting GroundedSAM with "cabinet." For the rest of the tasks, we annotate the task-relevant object locations. Note that since the positions of objects in the environments are fixed, we only need to annotate the position of the task-relevant objects once.

## A.3   Real Robot Experiment Setup

We use a 7-DoF Franka robot arm with a continuous joint-control action space at 15 Hz. A Zed 2 camera is positioned on the table's right edge, and only its RGB image stream—excluding depth information—is employed for data collection and policy learning. Another Zed mini camera is affixed to the robot's wrist. We encode the wrist image with DINO-v2 and pass the CLS token as an additional token to the policy during training. Operating under velocity control, our robot's action space encompasses a 6-DoF joint velocity and a singular dimension of the gripper action (open or close). Consequently, the policy produces 7D continuous actions.

## A.4   Additional Ablation Experiment for Real Robot Setup

Similar to the simulated experiments, we perform the ablation experiments Ours−multi-level where we remove object decomposition. Our main observation is that compared to this ablation, our method performs more robustly in more complicated tasks where identifying parts is crucial to the task's success. Especially in `Pout Water From Kettle into Pot`, where a firm and secure grasp is needed to pick up the kettle and precise location of the pot is needed, Ours−multi-level succeed 11 times in IND setup and only 6 times in OOD setup, proving that having the ability to identify and locate the parts greatly improves the task success rate in both IND and OOD cases. Full results are in Table 2.

| Task
Method | Eggplant-Sink | | Kettle-Stove | | Faucet | | Eggplant-Pot | | Water-Pot | | Overall | |
|---|---|---|---|---|---|---|---|---|---|---|---|---|
| # of Trials | 15 | 15 | 15 | 15 | 15 | 15 | 15 | 15 | 15 | 15 | 75 | 75 |
| | IND | OOD | IND | OOD | IND | OOD | IND | OOD | IND | OOD | IND | OOD |
| HODOR (Ours) | **14** | **12** | **13** | **12** | 12 | **8** | **13** | **8** | **12** | **9** | **64** | **49** |
| Ours−multi-level | 14 | 12 | 11 | 11 | 12 | 8 | 10 | 8 | 11 | 6 | 58 | 45 |

Table 2: IND and OOD BC Results on Real Robot Tasks. We report the number of success of each task out of 15 trials.

## A.5 LLM-based task-relevant object name identification

In this paper, we focus on evaluating short-horizon tasks, where natural task descriptions typically make the relevant objects immediately transparent. This is true for our tasks: [Put Eggplant in Sink, Place Kettle on Stove, Open Faucet, Put Eggplant in Pot, Pour Water from Kettle into Pot]. Identifying the right entities (e.g. "eggplant" and "sink") in these task names is very easy for modern LLMs, so we used very simple prompts: "Imagine you are a robot performing the task "⟨insert-task-name⟩", what are the task-relevant objects?" With GPT-4, this works with 100% accuracy across all our tasks. We include the results from GPT-4 for all tasks in Fig 9(a)-(e).

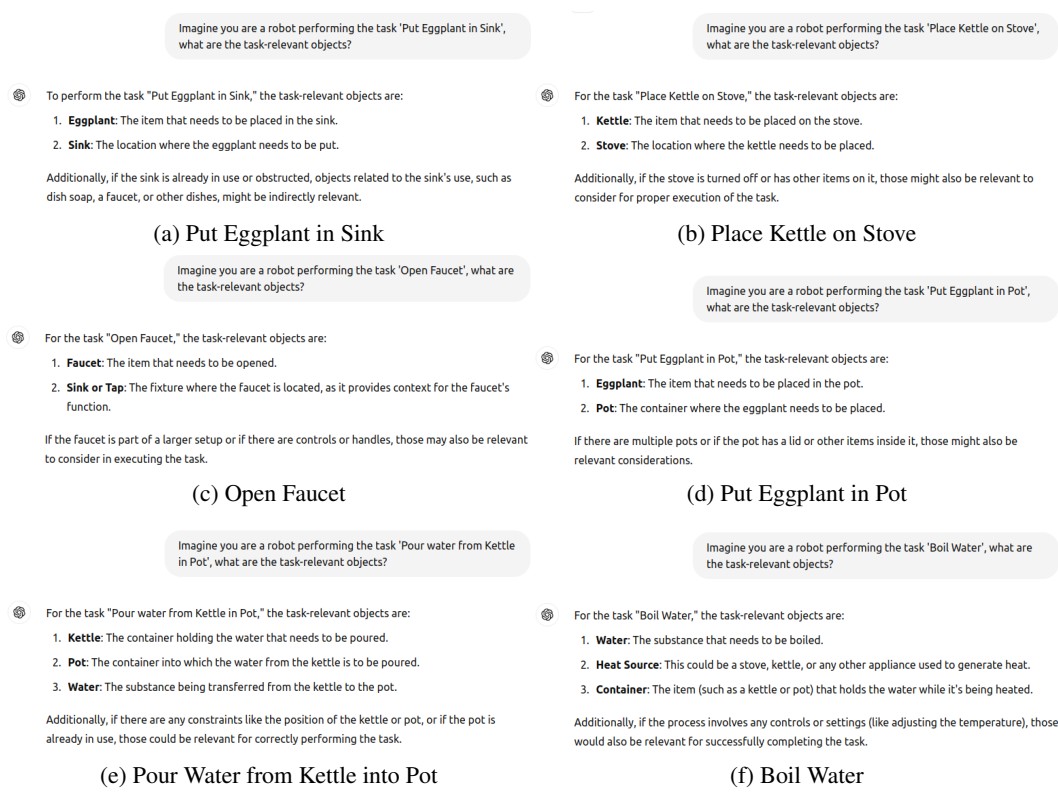

Figure 9: GPT-4 responses for task-relevant object name identification

## A.6 Technical Details for Generating Part Masks and Training

We first identify masks for the task-relevant objects by prompting GroundedSAM. Next, we ask SAM to generate all possible masks in the scene and keep the ones that have an intersection 0.5 with the task-relevant object masks. In this way, we generate a comprehensive list of part masks for every task-relevant object. We provide a visualization of objects and part masks for each real-

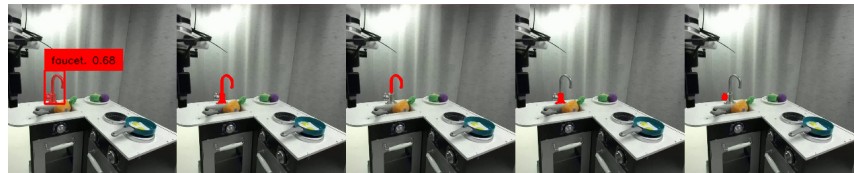

(a) Object-part segmentation masks for faucet.

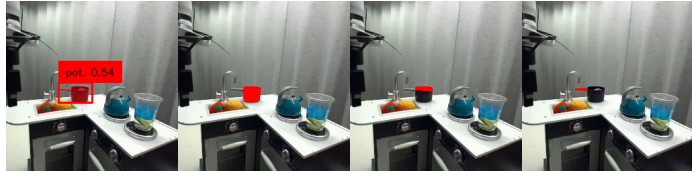

(b) Object-part segmentation masks for pot.

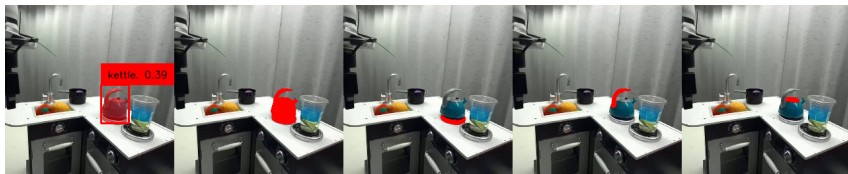

(c) Object-part segmentation masks for kettle.

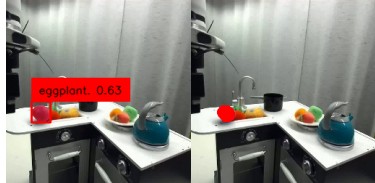

(d) Object-part segmentation masks for eggplant.

Figure 10: GroundedSAM results for detecting task-relevant objects in real-robot setting. The first image for each row shows GroundedSAM's detection and segmentation results. The following frames show the part masks for that task-relevant object.

robot task in Fig. 10. During training time, we precompute the segmentation masks and build the representation for each frame. So no pre-trained model is occupying the GPU during training time. We use one Nvidia 3090 GPU to train each mode and training one model roughly takes 6-8 hours.

## A.7 Limitations and Failure Case Analysis

While combining multiple pre-trained models might a priori carry the risk of error aggregation and brittleness compared to other approaches, we do not observe this to be the case empirically for HODOR in our extensive evaluations across 10 simulated and real-robot tasks. However, it is still valuable to understand how component failures affect task performance, and we present this analysis below.

**Failures of LLM-based task-relevant object name identification:** It is plausible that natural task names sometimes do not reveal the relevant objects so easily, particularly for long-horizon / high-level tasks. To study what might happen in such settings, we have now paraphrased the "Place Kettle on Stove" task to "Boil Water". GPT-4 still succeeds in generating the suggestions that contain the correct object names, as shown in Fig. 9(f). GroundedSAM will still return the same results in this case, since it will simply return an empty list and if the suggested objects are not in the scene. For still more challenging settings e.g. a long multi-step task like "make a sandwich", careful prompt engineering may be necessary, together with the use of a VLM rather than a pure language model.

**Failures of segmenting/tracking the objects of interest:** Recall that we prompt Grounded-SAM with the list of objects generated by GPT-4 to segment those objects the first frame, and afterwards propagate the masks of task-relevant objects to all following frames using XMem tracker. In our experiments, first frame segmentation failures are rare: this happened in 1 out of 75 of the real-robot evaluation episodes, and the task policy fails when this happens. This failure case is visualized in Fig. 11. Tracking failures are more common: task-relevant objects or their parts may be lost during the episode due to occlusions or failures in the tracking model, as illustrated in Fig. 12. While common, this does not usually result in task failure. Recall that our representation is structured as a coarse-to-fine representation, and the coarsest level is a representation of the full scene, independent of segmentation/tracking failures. In this sense, HODOR representations have some built-in redundancy that can aid in robustness. Occasional tracking failures are typically easily handled, and sometimes even segmentation failures.

Our approach could be further robustified against first frame segmentation failures by running segmentation on multiple keyframes throughout the episode, e.g. once every k frames, rather than only on the first frame. This implementation-level improvement might improve HODOR performance in more challenging tasks in the future, particularly if objects of interest are not visible in the first frame. To sum up, while each component model in the HODOR system does indeed offer a point of failure, the overall system is often capable of correcting for these errors, and/or these points of failure are rare in the types of tasks most frequently studied in imitation learning for manipulation today. This is backed up by the fact that HODOR outperforms all baselines in our extensive evaluations

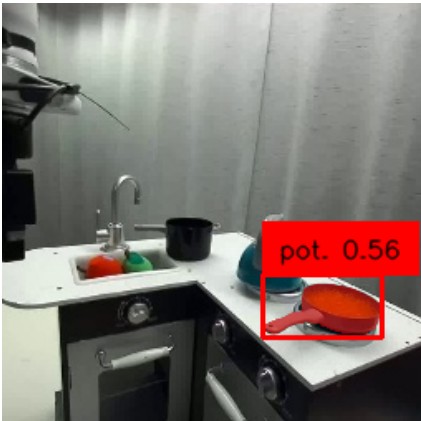

Figure 11: GroundedSAM error. In this task, the task-relevant object should be the black pot in the background as well as the kettle. However GroundedSAM outputs a false positive mask for the pan in front.

## A.8 Additional Ablation Experiments

In addition to the baselines in Sec 5.1, we show results on two additional baselines: 1) **dense DINO** is the variant of **Ours-multi-level**, but instead of using the last-layer CLS token as the scene-level representation, we use the concatenation of the dense DINOv2 features. 2) **task-conditioned dense DINO**(tc dense DINO) is a baseline that, instead of selecting task-relevant information using GroundedSAM, we concatenate a CLIP embedding on the task description with the dense DINOv2 representations, and use the same number of self-attention layers as in HODOR to process this dense representation. AS Fig. 13 shows, both of these baselines performs poorly—much worse than HODOR on all five Franka kitchen tasks, ranking near the bottom of the baselines on most tasks. This is not surprising for several reasons. First, using dense DINO features instead of the CLS token would increase the policy complexity. Assuming a patch size of 16 by 16 and an image size of 224 by 224, concatenating the dense features would result in 14*14=196 feature tokens instead of 1 (CLS) token. This increases the policy complexity and thus decreases the sample efficiency: we

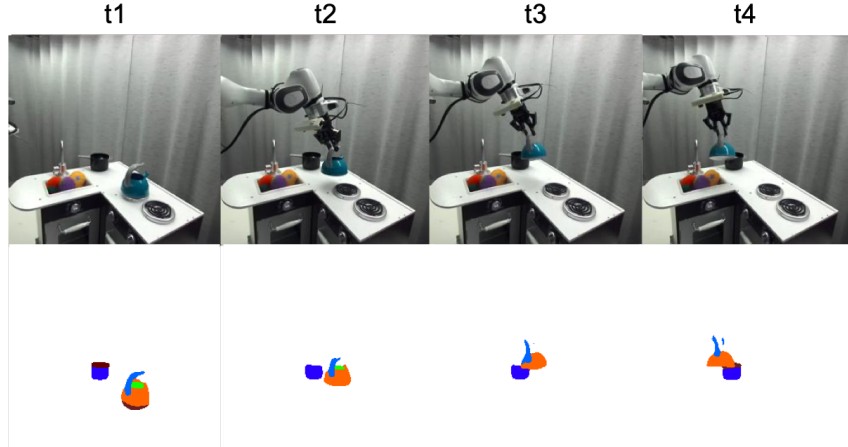

Figure 12: Tracking errors due to occlusions are fairly common during policy execution. In this evaluation episode from the Pour Water in Pot task, XMem loses track of the kettle lid after the robot picks up the kettle, and the handle becomes heavily occluded.

would need a lot more data to train this dense DINO policy. Second, since CLIP features are not trained to be aligned with DINO, conditioning on these CLIP features won't help the policy focus on task-relevant features. In contrast, HODOR focuses on task-relevant objects to form a more compact yet sufficient representation, resulting in more sample-efficient policy learning.

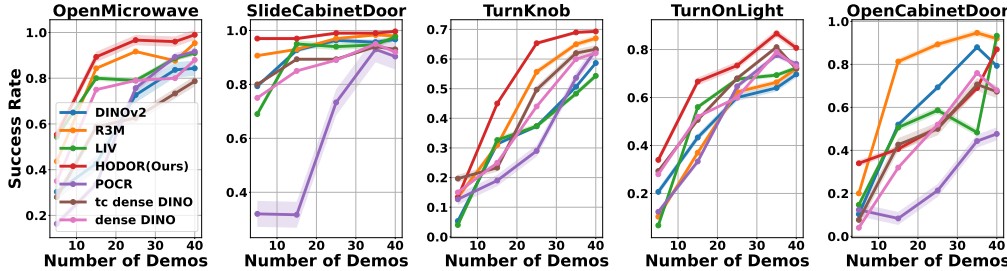

Figure 13: Results for additional baselines: dense DINO and tc dense DINO. Both of them are perform poorly on FrankaKitchen tasks. This is likely due to the lack of sample efficiency when using dense DINO features.

## A.9 Differences between Ours-Multilevel and POCR

**POCR** is not just a 1-level implementation of HODOR: there are subtle differences that are nevertheless important. 1) POCR requires that the set of object slots be ordered in the same way across all episodes for a task. This causes brittleness when an object is occluded or less visible in some episodes relative to others. HODOR instead orders object slots through tracking only within an episode, not across them. 2) POCR constructs object slot embeddings by using each object mask to create a masked object-specific version of the RGB image. before encoding that image with LIV. For a small object, the masked image is almost entirely blank, with only a few pixels occupied by the object; the LIV encoding can fail to capture sufficient details of the object. To avoid such failure points, HODOR uses the masks directly to pool DINO features within each object region. These differences are proven to be important, as when we compare Fig 5 with Fig 6, Ours-multilevel outperforms **POCR** in most cases.

