# OpenReview forum: "Task-Oriented Hierarchical Object Decomposition for Visuomotor Control"
_robot-learning.org/CoRL/2024/Conference — CoRL 2024_

### Official Review · Reviewer_9qHC · 2024-07-01
**A combination of large pre-trained models for manipulation tasks**

**Originality:** 2
**Technical Quality:** 3
**Clarity Of Presentation:** 3
**Potential Impact:** 2
**Recommendation:** 3
**Confidence:** 4

**Review:**

**Strengths**:
* The idea of providing part-objects, or multi-level features of objects is interesting and intuitive.
* Clear and easy to follow.
* Competitive or slightly better results than previous works, seems to shine in low-data regimes.
* Real robot experiments.
* Interesting OOD experiments.
* Broad related-work section.

**Weaknesses**:
* Weak generalization capabilities to unseen task-relevant objects.
* A large pipeline of large pre-trained models – the requirement of having all these models in memory to complete a task sounds very impractical. Even if we load them sequentially, it seems that there is a long pre-processing computation step to solve a given task.
* Strong reliance on supervised models (mainly SAM) - this is obviously a limitation for all models that rely on these supervised models which may harm generalization in case objects (think about deformable objects for example) are not recognized by SAM (which is in accordance with lines 197-200: “...In practice, prompting Grounded SAM with the object name in the simulated environment does not reliably identify the objects due to the fact that objects in the simulated environment could look ambiguous ”). I find this a significant limitation of the model since it requires more human intervention (e.g., manually labeling data~line 201).
* Moderate novelty: from Figure 5, it seems that the additional level of granularity provides very small gains.
* At the time of reviewing this paper, the project site (https://sites.google.com/view/hodor-corl24) was empty. I watched the video in the supplementary material, but I believe it is important to accompany the results of this paper with a more structured format, like a project page.

**Limitations**: Limitations are not addressed. I would suggest the authors add a paragraph on the limitations of the method, and there is no short of them as I raised in my review.

**Minor Comments**:
* Lines 203-204: the sentences are not coherent.
* You use the abbreviation “BC”, which I assume stands for “Behavioral Cloning”, but you never make it explicit throughout the text even once.

**Quality Of The Limitations Section:**

1

**Questions For Rebuttal:**

* Concatenation vs. [CLS] token for pooling the output of the Transformer: I understand the memory reasoning for the real-robot experiments, but do you have actual ablation of the pooling type (even if simulation)?
* Missing technical details: [lines 150-151] “...SAM to obtain all the segmentation masks at different levels of granularity..”: there is no explanation, even in the appendix, of what exactly is done in this step. Can the authors explain what are the “different levels of granularity” and what is done in practice? Objects can differ in size which might affect how they are further decomposed to parts. As this is the main contribution of the paper, I’d expect a deeper discussion/explanation of this.
* R3M seems very competitive to HODOR, and significantly outperforms it for the “Open Cabinet Door” task. Why do you think R3M fares better than HODOR in this case?
* Additional training details: what kind of hardware and computational resources were used to train the models? How long does it take? How do you deal with the memory required for all the large pre-trained models?
* Why is “skill-chaining” an interesting feature? It seems that the robot state is reset after each task, so it is like solving each task separately. If the authors argue that the previous execution altered the environment state, this is just like testing if the method generalizes to the same task with different initial environment states.
* Do the authors plan to release an open-source code?

**Robotics Focus:**

4

**Summary Of Paper:**

The paper proposes a method (HODOR) that is built upon the idea of several previous works (and notably POCR) – a pipeline of several large pre-trained models (SAM, XMem, GPT, DINOv2) to supply a decomposed representation of the scene for the downstream task of object manipulation via imitation learning. The idea is to provide a hierarchical tree of object parts instead of just the entire segmented object. The relevant entities are extracted by prompting a large network to identify objects of interest in a textual description of the task.

**Summary Of Recommendation:**

Overall, I see very moderate contribution and novelty in this paper. I have raised several concerns in the “Weaknesses” and “Questions” parts of my review that I would like the authors to address. As such, my current recommendation is “Weak Reject”, but I’m willing to increase my score given convincing answers to my concerns and the concerns that might be raised by the other reviewers. **Post-Rebuttal**: I have read the other reviews and it seems that most of the reviewers raised the same concerns regarding the limitations of this method of relying on a myriad of pre-trained models and the moderate novelty. The paper is well-positioned in the "using large pre-trained models for robotics" literature, and while I still have my concerns regarding this pipeline, I appreciate the authors' response and increase my score, but I will not argue against rejecting it if it is the general consensus amongst the other reviewers.

---

### Official Review · Reviewer_fKHY · 2024-07-18
**novel approach to representing the state of a BC policy. This approach focuses on a structured representation of a scene based on objects and object parts allowing for sample efficient learning and ood generalisation. Interesting idea.**

**Originality:** 4
**Technical Quality:** 4
**Clarity Of Presentation:** 3
**Potential Impact:** 3
**Recommendation:** 4
**Confidence:** 5

**Review:**

Quality
The quality of the work is good and presents an interesting and structured approach to representation learning for behaviour cloning

Clarity
The paper is generally clear and methodology is well explained.

Originality
The idea of object centric decomposition within a structured heirarchy is novel and provides better interpretability that prior works.

Significance
This work addresses an important challenge in behaviour cloning where learning directly from images poses several challenges and can require alot of data even for simple generalisation to distractors and task irrelevant entities. The representation is also very interpretable which is great for understanding where the policy could fail.

Strengths
- Simple and structured approach to addressing the sample efficiency in robot policy learning.
- The representation in interpretable.
- The object centric approach is important to structure complex problems into more manageable sub-problems.

Weaknesses

- Do you have to keep segmenting the parts at each step across the policy? What is the computational overhead for this?
- The method is reliant on the vision system having good observability of the scene objects and thier parts. If there are occlusions we see how this could deteriorate performance. If the policy is operating on the kettle for example and as it moves the handle gets obscured from the camera does this mean that there is no segment for the handle representation mid execution? How does this impact policy performance?  Do you assume that all parts and the full representation for an object remains consistent across the policy or can some parts disappear? - Some results to show this would be good.
- How sensitive is your approach to poor segmentation?

**Quality Of The Limitations Section:**

1

**Questions For Rebuttal:**

I summarise the questions raised above below:

- Do you have to keep segmenting the parts at each step across the policy execution?
- What is the computational overhead for segmentation at each step?
- How does the method handle scenarios where parts of objects are occluded during task execution?
- If an object's part, such as the handle of a kettle, gets obscured from the camera mid-execution, does the segment for that part disappear? How does this impact policy performance?
- Do you assume that all parts and the full representation for an object remain consistent across the policy execution, or can some parts disappear? If parts can disappear, how is this managed?
- How sensitive is your approach to poor segmentation?

**Robotics Focus:**

4

**Summary Of Paper:**

A novel framework for robot visuomotor control by creating task-specific visual representations. Organizes scene entities hierarchically into objects and parts, allowing robots to focus on task-relevant information. This method improves sample efficiency and performance in complex scenes by filtering out irrelevant details based on language-based task descriptions. Key contributions include a multi-level scene decomposition and a task-conditioned entity selection.

**Summary Of Recommendation:**

Good paper, some additional results to support the robustness of this approach would be good to help support the applicability of the method to more general settings. I believe the idea is good and the paper could be improved based on the questions above.

---

### Official Review · Reviewer_cTv2 · 2024-07-20
**Review for "Task-Oriented Hierarchical Object Decomposition for Visuomotor Control"**

**Originality:** 2
**Technical Quality:** 3
**Clarity Of Presentation:** 3
**Potential Impact:** 2
**Recommendation:** 3
**Confidence:** 3

**Review:**

# Strength
1. The paper is showing an advantage of using more task-oriented representations that focus only on the object mentioned in the task.
2. In addition, the paper shows that representing only object, but also their parts found by SAM also has a positive effect on the performance.

# Weaknesses
1. While interesting, such filtering of objects has many limitations that are not discussed in the paper. Some of them are:

    - If an object is not visible in the first frame, it would not be segmented and thus would not appear in the scene representation throughout the whole video. This may cause problems when we want to find objects that are occluded or inside other objects. For example, “take the eggplant from the microwave” would have no representation of the eggplant if the microwave is opened in the middle of the trajectory.

    - If the task does not explicitly mention an object, it would be filtered out. For example, a task like “clean the table” would filter out the cleaning tissue, as it is not explicitly mentioned in the task.

    - If there are several objects of the same type, it is not clear if such filtering would work. For example, in the case of two eggplants near each other, how would the reference “take the left eggplant” work? Would the representation include both eggplants, or only the correct one?

2. A simple but important baseline seems to be missing. In contrast to selecting objects by GroundingSAM, one can use dense DINOv2 representations processed with Self-Attention conditioned on the embedding of the task description or some learnable representation of the task. In contrast to the global representation from DINO, such representation is more robust to changes in the scene and can be used as a good baseline to show if object-centric representation is better than processing dense patch representation with SA.
3. Some additional work [1, 2] on object-centric representation in real world may be relevant


[1] [DINOSAUR: Bridging the Gap to Real-World Object-Centric Learning, ICLR2023](https://openreview.net/pdf?id=b9tUk-f_aG)

[2] [Unsupervised Open-Vocabulary Object Localization in Videos, ICCV2023](https://arxiv.org/pdf/2309.09858)

**Quality Of The Limitations Section:**

2

**Questions For Rebuttal:**

# Questions:

1. Parts are defined heuristically as anything that intersects with the object sufficiently (>0.5). It is not clear why IoU is used. The definition of a part is that the intersection is the size of the smaller mask, whereas the union is the size of the larger mask. Thus, this criterion (with some thresholds, e.g., [0.9, 1.1] of the original size) seems like a natural definition of a “part.”
2. Why is the object-level “positional” encoding unique for each object? How is it determined how many of these encodings would be learned? Are they based on the language description of the objects?

**Robotics Focus:**

4

**Summary Of Paper:**

Paper proposes 3 levels object-centric representation for more efficient and robust BC. The paper uses ChatGPT + GroundingSAM to filter non-important object representations and represent the scene with only necessary objects for a particular language conditioned task

**Summary Of Recommendation:**

While promissing current version of the paper is missing some imporant simple baselines and are not adressing some of the limitations.

---

### Official Review · Reviewer_U4ZG · 2024-07-21

**Originality:** 3
**Technical Quality:** 4
**Clarity Of Presentation:** 4
**Potential Impact:** 3
**Recommendation:** 2
**Confidence:** 5

**Review:**

This paper presents an approach that first uses LLMs to extract task-relevant object and part names (e.g., microwave, door, etc. in the task of "opening a microwave"), and then learn policies for the tasks based on DINO-encoded relevant object/part features.

The paper focuses on the evaluation of data efficiency: for most tasks shown in Figure 3, given a sufficient amount of data, most models (including baselines) tend to perform decently. The new model, HODOR, shows stronger performance especially when the amount of training data is small.

One of the main concerns of the paper is its applicability to realistic and complex scenarios, coming from three sources.

First, based on my understanding, this paper crucially relies on the competence of several large language and vision-language models, such as GPT-4 for suggesting task-relevant objects and parts, and Grounding SAM and XMEM for detecting and tracking objects. While it is indeed true that these "foundation models" are performing better and better, the authors should at least discuss the limitations of these models and provide failure examples (for example, what will happen if the GPT-4 model fails to identify a task-relevant object? what if it suggests objects that do not appear in the scene? what if the object detection model fails? what if the tracking model fails?).

Second, and more importantly, such "task-relevant" object identification is fundamentally limited in densely packed scenes, where in order to move a particular object/part, several other obstacles should be taken into consideration (at least for collision-avoidance purposes). The authors showed some examples in their supplementary material but I think this is a very important limitation of the current system and should be clearly discussed in the experiments section (besides just being mentioned in the supplementary video).

Third, identifying task-relevant objects solely from a single text description of the task, sounds very brittle to me. For example, considering the opening-microwave example, there can be many, many different types of microwaves that require the usage of different objects/parts to open (e.g., with buttons, with nobs, with handles, with one doors/two doors). In the current form of the system, I do not think there is any feedback to the LLM module to refine the selection of relevant objects (at test time). Therefore, the authors should evaluate their system on a wide variety of objects and scenes (e.g., different types of microwaves, from, for example, https://maniskill2.github.io/), in order to verify the robustness of the system. Or, they should discuss this limitation clearly **in** the experimental section.

**Quality Of The Limitations Section:**

2

**Questions For Rebuttal:**

1. In Figure 4 and Figure 5, is POCR essentially Ours-TC-Multilevel? Based on my understanding, POCR = object-level representation only, and no task conditioning. Then, the results are quite interesting because the performance comparison becomes:

Ours > Ours-multilevel > Ours-TC-multilevel > Ours-TC.

Is this correct? Or are there any other differences between Ours-TC-Multilevel and POCR?

2. Can you provide details of the LLM promoting? Specifically, can you provide the raw prompts, queries, and LLM responses? How are these prompts built and how do you handle failures/stochasticity in LLM-generated outputs?

3. There is related work on part-based manipulation policy learning that should be cited and discussed.

One-Shot Transfer of Affordance Regions? AffCorrs!: https://arxiv.org/pdf/2209.07147
Composable Part-Based Manipulation: https://arxiv.org/pdf/2405.05876
PartManip: Learning Cross-Category Generalizable Part Manipulation Policy from Point Cloud Observations: https://arxiv.org/abs/2303.16958

**Robotics Focus:**

4

**Summary Of Paper:**

This paper presents a novel approach based on hierarchical object-part based representations for imitation learning in robotic manipulation tasks.

**Summary Of Recommendation:**

See my questions and weakness reviews. I am happy to raise my score if the limitations of the system are clearly studied (experimentally) and discussed in revisions.

---

### Author Rebuttal · Authors · 2024-08-10

Updated paper and appendix.

---

### Decision · Program_Chairs · 2024-09-04

**Decision:**

Accept

**Comment:**

The reviewers broadly agree that it is important to find representations that enable learning from a small number of examples.  The proposed method is sensible, but there is concern about its generality, in the face of situations where other objects are important (e.g., something needs to be moved out of the way) or there is substantial occlusion.  An additional concern is the robustness of the system:  what happens in the face of (inevitable) failures of the component foundation models?  In the final revision, it will be important to be as clear as possible about the degree to which this method generalizes (to different layouts, sets of parts, especially when substantially different action sequences are required) and to talk about strategies for improving in that direction.